# From Safety Net to Point of No Return—Are Small-Scale Inland Fisheries Reaching Their Limits?

Pina Lena Lammers, Torsten Richter and Jasmin Mantilla-Contreras *

Ecology and Environmental Education Group, Institute of Biology and Chemistry, University of Hildesheim, Universitätsplatz 1, 31141 Hildesheim, Germany; pina.lammers@posteo.de (P.L.L.);
richtert@uni-hildesheim.de (T.R.)
* Correspondence: mantilla@uni-hildesheim.de

**Abstract:** Small-scale inland fisheries (SSIF) are a livelihood opportunity for millions of people in developing countries. Understanding the economic, ecological, political and social impacts fishers are coping with can clarify weaknesses and challenges in the fishery management. Using the SSIF at Lake Alaotra, Madagascar, as an example, we analyzed the development and fishers' perception of, and adaptation strategies to, change. We surveyed fish catches to assess the state of fish stocks and conducted interviews to understand fishers' livelihood, problems, behavior and attitudes. Our results show that the fishery sector of Lake Alaotra has grown dramatically although fish catches have fallen sharply. Changes in species composition and low reproduction rates reflect the fishing pressure. A point of no return seems near, as decreasing agricultural yields force farmers to enter the fishery sector as a form of livelihood diversification. Lake Alaotra reflects an alarming trend which can already be seen in many regions of the world and may affect a growing number in the near future. The Alaotran fisheries demonstrate that SSIF's ability to provide livelihood alternatives under conditions of insecurity will become increasingly important. It further highlights that the identification of ongoing livelihood dynamics in order to disclose possible poverty trap mechanisms and to understand fisheries' current function is essential for sustainable management.

**Keywords:** small-scale fisheries; livelihood opportunity; poverty trap; welfare function; developing countries; Madagascar; Africa

## 1. Introduction

Small-scale inland fisheries (SSIF) are a growing livelihood opportunity for people living next to water bodies in developing countries [1–3]. In many local communities in the developing world, fish is the primary source of protein, being critically important for human health and child development [2]. Today, over 85 percent of the world's capture fishers are employed in small-scale fisheries, providing food and nutrition security as well as cash income to local communities in the developing world [4,5]. In developing countries, SSIF and related activities provide employment to 60 million people [6]. However, SSIF, as fisheries in general, are increasingly facing challenges, e.g., growing demand and environmental degradation, especially water pollution [7] and climate change [8], that may endanger fishery-based livelihoods worldwide and calls for particular attention in decision-making processes and the adapted management of resources. Nevertheless, small-scale fisheries (SSF) are poorly integrated in regional and national policy as well as scientific research [6,9,10]. Their decentralized, diverse and diffuse nature makes it difficult to give a general picture of SSIF and to measure their impact. As a result, the inland fishery sector, whose production is mainly based on small-scale activities, is often dismissed as "backward, informal and marginal" [4]. This neglect is fueled by several interrelations, which are often seen as one-sided or simplified: First, the geographical isolation of

fisherfolks, mostly living in marginal or remote areas with poor infrastructure, is usually seen as an impediment for development [11]. Second, catches from small-scale fisheries often serve domestic consumption or enter the market through informal ways. Therefore, the contribution to national economies is largely underestimated, and small-scale fisheries are often seen to offer little potential for development, although more than half of the catch in developing countries is caught by the small-scale sector [6]. Third, the development of water and water-related resources, like dams for hydropower, is commonly considered contrary to the aspirations and needs of fishers to their social and ecological environment [11,12]. Forth, capture fisheries are largely seen as being doomed to decline due to their open access nature, facing pressure from population growth, land use changes, infrastructure development and environmental changes [12], leading inevitably to "Malthusian overfishing" [13] and what Hardin [14] called the "Tragedy of the Commons". Based on its open access, fisheries are, according to Béné, often reduced to a "last resort activity" for poor people [15], and fishers are referred to as the "the poorest of the poor" [15], since the use of common property is said to lead inevitably to a low level of resources and low income [15]. However, this one-sided perspective ignores the fact that fishers are not necessarily poor because they are fishers, but they are fishers because they are poor (e.g., landless). Fisheries therefore should also be regarded as a "safety net", and their open access nature, though it does not hold true for all small-scale fisheries [15], considered positive [16]. As a consequence of common simplistic and reductionist conclusions that underestimate their potential, inland fisheries often lose out to other sectors and interests [17]. Besides the political marginalization, this becomes apparent through an "inadequate financial, institutional, and scientific support for small-scale fisheries" and has "further obscured evidence about the contribution of small-scale fishing communities to conservation, [ . . . ] poverty alleviation, social well-being and resilience, and cultural heritage", as Chuenpagdee [8] summarized (p. 22).

Thus, little is known about the drivers of resource exploitation and adaptation strategies of local small-scale inland fishers to environmental and demographic changes. Studies on changes in fisheries have so far focused mainly on marine ecosystems [8,18,19]. Understanding the larger economic, ecological, political and social impacts local small-scale fishers are coping with as well as their vulnerability and perception of change can clarify weaknesses, gaps and challenges in current fisheries management and help decision makers, including fishers themselves, to develop locally and regionally appropriate adaptation strategies [20,21].

The small-scale fishery at Lake Alaotra, in Madagascar, is currently confronted with major environmental and demographic changes and functions therefore as a case study to investigate fishers' resilience and adaptation strategies: the region has undergone significant changes in the past. The human population has increased fivefold during the last 50 years to more than half a million people [22,23]. The drastic population growth had detrimental effects on the entire ecosystem: the low social resilience of the rural population due to high dependency on natural resources entails high human demands on freshwater resources and agricultural land and has resulted in overfishing, marsh burning and the subsequent conversion into rice fields (cultivation of "*vary jebo*" rice during the dry season), leading to wetland fragmentation. Further, pollution, introduced fish and plant species have altered the freshwater species diversity and composition. The transformation of water bodies and streams for agricultural irrigation, as well as the increasing sedimentation, have already reduced the lake to 20–30% of its original size [23,24]. Overfishing and environmental degradation have resulted in declining fish catches and decreased crop production of approximately 40% during the last years and led to additional pressure on the remaining natural resources [22,24].

Using the SSIF at Lake Alaotra, this study analyzes the current challenges small-scale fishers are facing as well as their perception of and vulnerability to change. We will answer the following questions: Which changes do small-scale inland fishers feel exposed to? What are the drivers of change? How do fishers adapt to those changes and what are the implications?

We analyzed social, environmental and economic changes and the respective consequences that may similarly affect small-scale fishers in rural regions in tropical countries all over the world.

## 2. Materials and Methods

### 2.1. Study Area

Lake Alaotra in northeast Madagascar (E 048°26′, S1 7°31′) is the country's largest freshwater lake and the base for the most important inland fisheries [25]. The shallow lake comprises 20,000 ha of open water and reaches a maximum depth of 4 m. It is surrounded by 23,000 ha of freshwater marshes and 120,000 ha of rice fields [22,26] (Figure 1). The region is characterized by a warm rainy season (November to March) with high precipitations (900 to 1250 mm on average, with a maximum of 250 mm in January), a rising lake water level and a cold dry season (April to October). The annual mean temperature is 20.6 °C and ranges from 11 °C in July to 28 °C in January [27]. The main sources of income are SSIF and rice cultivation [26]. The overall economic productivity of the region lifts the daily income (2.5 USD to 5.0 USD) well above the national average (78% of the Malagasy population have a daily income below the poverty line of 1.90 USD) and causes strong immigration [28,29]. The wetland complex is internationally recognized as a Ramsar site since 2003 and as a new protected area (Nouvelle Aire Protégée, NAP) by the government of Madagascar since 2007 [26]. Poor enforcement and low compliance, however, have led to Lake Alaotra's wetland complex being widely regarded as a "paper park" [30].

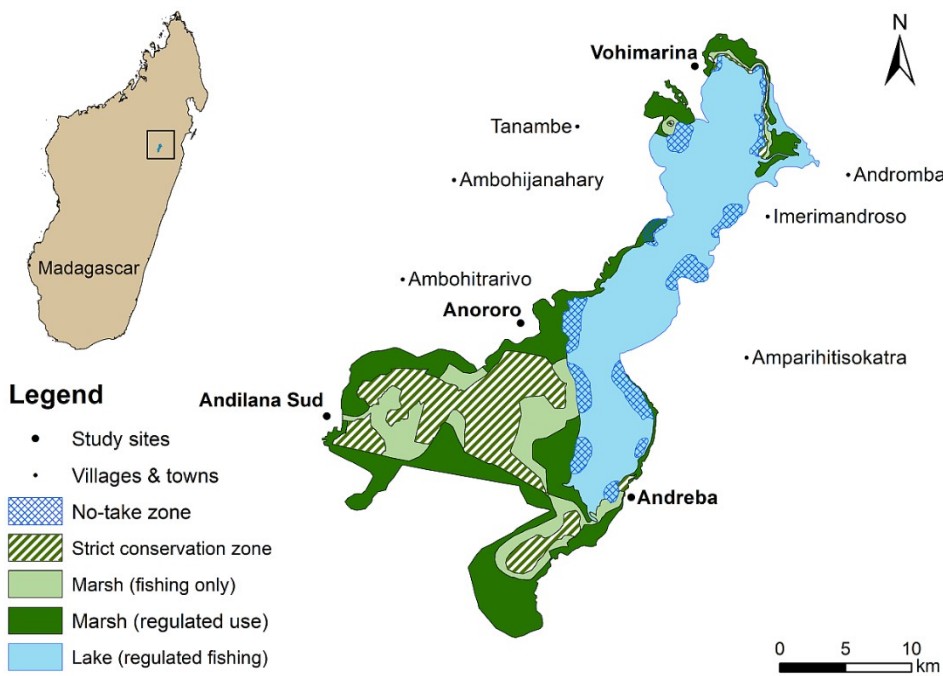

**Figure 1.** Map of Lake Alaotra and its wetland management zonation. Fishing activity is prohibited in strict conservation zones in the center of the marshes and in no-take zones around the lake-edge modified from [30,31].

Today, the vast majority of fish species in catches at Lake Alaotra are introduced (Table 1). The fishery management of Lake Alaotra is enforced through the Fishery Service (*Service Régional de la Pêche et des Ressources Halieutiques*), based in Ambatondrazaka. Another authority, the Federation of Fishers (*Fédération des Pêcheurs*) is supposed to represent the local fishers and to implement, monitor and develop fishery regulations directly on-site. However, the Federation is suffering from a lack of confidence by locals, due to few interactions with local fishers and a lack of commitment in management [30]. Since 1998, fisheries at Lake Alaotra are regulated by gear restrictions and minimum fish size limits. A temporal fishing closure (15 November to 15 January) and spatial fishing closures (fishing is prohibited in no-take zones near the lake edge and strict conservation zones in the center

of the marshes), implemented in 2001 and 2006, respectively, aimed to increase the effectiveness of management interventions. The two-month fishery closure aims to allow for the recovery of the fish stocks, while no-take zones are intended to permanently protect fish spawning sites [30–32].

**Table 1.** Introduced and native fish species (predominant fish species in bold letters) in catches at lake Alaotra.

| Fish Species | Distribution Status Class |
|---|---|
| Nile tilapia (*Oreochromis niloticus niloticus*) | Introduced |
| Redbreast tilapia (*Tilapia rendalli*) | Introduced |
| Tilapia hybrids | Introduced |
| Blotched snakehead (*Channa maculata*) | Introduced |
| Common carp (*Cyprinus carpio carpio*) | Introduced |
| Goldfish (*Carassius auratus auratus*) | Introduced |
| Eastern mosquitofish (*Gambusia holbrooki*) | Introduced |
| Black bass (*Micropterus salmoides*) | Introduced |
| Mozambique tilapia (*Oreochromis mossambicus*) | Introduced |
| Indonesian short-finned eel (*Anguilla bicolor bicolor*) | Native |
| African longfin eel (*Anguilla mossambica*) | Native |
| Madagascar rainbowfish (*Rheocles alaotrensis*) | Native |

Sources: [30,31].

## 2.2. Data Collection

Fish surveys were conducted in 2013 and 2014 to measure fish catches regarding catch weight (g), number of caught fishes (*n*), fish species (or genus, in the case of tilapia and eel) and size of caught fishes (cm). Fishes were assigned to three size categories (A = < 13 cm, referring to the minimum permissible length of *Tilapia* in catches; B = 13–20 cm; C = > 20 cm). A total of 859 catches were recorded at different times of the year (March to April 2013, January, June, September to October 2014) at four sites: Andreba, Anororo, Vohimarina and Andilana Sud (Figure 1). Locations encompass a range of community types, from small village to bigger communes (Inhabitants: Anororo, 8000; Andreba, 5000; Vohimarina 500; Andilana Sud, 3900; pers. comm. with village chiefs 2016; [30]).

To gain a better understanding of fishers' attitudes and behaviors, data of fish catches were supplemented with information about fishers' everyday working life, problems and perceptions through structured interviews (see Table S1). Interviews, consisting of 39 closed and 2 open questions (indicated with Q and the respective number of the question in the interview, e.g., Q1), were conducted in March 2017. 117 fishers were interviewed, 28 to 30 in each of the four sites. Interviews were conducted by a local Malagasy research assistant and took place at the local harbor. The categories for closed questions were developed based on the respondents of a preliminary study conducted in Andreba with 44 fishers. Preselected categories minimized the required time for conducting interviews in order to increase fishers' probability and willingness to participate. Interviews lasted app. 30 min and addressed issues about their work as fishermen (e.g., gear specification, daily working routine, income and livelihood diversification) as well as their perception about changes, problems and the future of Lake Alaotra's fisheries.

In order to draw a general picture of the Lake Alaotran fisheries since their expansion in the middle of the 20th century, additional data about yearly fish production and number of fishers were collected from the literature and from the regional directorate of fish resources and fisheries (*Direction Régionale des Ressources Halieutiques et de la Pêche*) in 2017.

### 2.3. Data Analyses

Data from fish catches (*n* = 859) were analyzed jointly for all sites and time periods. Overall, 117 interviews were conducted and later used for data analyses. Interviewed fishers (*n* = 117) at Lake Alaotra were all male, on average 40 years old (age 17 to 71 years) and had 17 years (1 to 51 years) of experience in their work as fishermen (Q1,2,4). Interviews contained questions generating qualitative and quantitative data. Qualitative data were analyzed by using qualitative content analysis [33].

The average intercoder reliability of 99% (*n* = 2) was calculated following Holsti [34] and ranged between 97% and 100%. Coding discrepancies were resolved through discussion before the analysis continued. Literal interview excerpts were indicated by an interview identifier made up of the sites, name and consecutive interview number (e.g., VF15). As multiple answers were allowed, the sum of response percentages could exceed 100%. Given percentages were calculated for the 117 respondents. In the case of subgroups, the exact sample size was indicated in brackets. All results from the interviews can be found in Table S2.

The income data obtained from the structured interviews were given in US Dollars (USD), using the exchange rate of March 2017 (MGA/USD = 3177), the time period when the interviews took place. Since data from fish surveys and interviews did not follow a normal distribution, average values were given as median (range).

## 3. Results

### 3.1. Evolution of the Alaotran Fisheries

The Lake Alaotran fisheries show, since their expansion in the mid 20th century, an opposite development: while fish catches (in tons) have fallen sharply since the 1960s, the fishery sector (number of fishers) is growing continuously, especially since 1975 (Figure 2). The maximum yield of the Alaotran fisheries was reached in the 1960s, with 4000 tons of fish per year allocated to around 1000 fishers. The latest available official data from 2011 show that annual fish catch has dropped to less than 1000 tons per year, while the number of fishers has increased more than tenfold, up to 12,000. Non-official data on fish catches up to 2016 indicate a stabilization at the low level reached in 2011 [35].

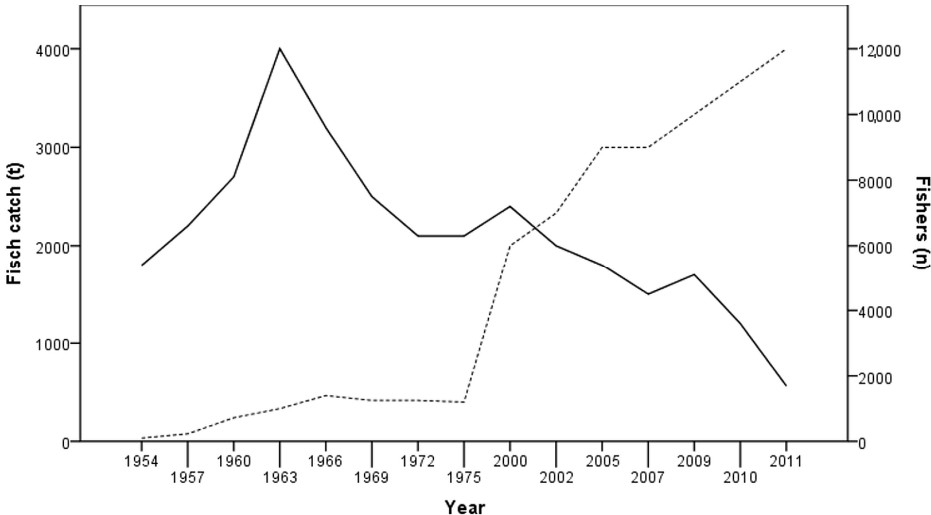

**Figure 2.** Annual fish catches and number of fishers in the Alaotra basin during the past 60 years. Solid line = annual fish catches in tonss; dotted line = number of fishers. Sources: [22,35,36].

The high pressure on the Alaotran fish stocks is reflected by our data from fish surveys: 80% of the caught fishes (*n* = 859 fish catches, *n* = 57,077 individuals) had a length of less than 13 cm (Category A). Only a small proportion (3%) reached the size category C with a length of more than

20 cm (Table 2). The species composition of fish catches was poor and consisted of seven species, or rather genera: *Tilapia* spp., *Channa maculata*, *Cyprinus carpio carpio*, *Micropterus salmoides*, *Carassius auratus*, *Anguilla* spp., *Awaous aenofuscus*: catches were dominated (*n* = proportion by number of individuals, w = proportion by catch weight) by *Tilapia* species (*n* = 94%, w = 82%) and, even though to a smaller extent, the invasive snakehead *Channa maculata* (*n* = 3%, w = 9%), while all other species account only for a small proportion (*n* = 1%, w ≤ 3%) of the catches (Table 2).

**Table 2.** Proportion of fish species (individuals and weight) and their size categories in total catch (*n* = 859 fish catches).

| | Size category (lenght) | | |
|---|---|---|---|
| | A (> 13 cm) | B (13 to 20 cm) | C (> 20 cm) |
| Number of individuals (n) | 45444 | 9927 | 1707 |
| Percentage (%) | 80 | 17 | 3.0 |
| Fish species (distribution status class) | Catch weight (kg) | Percentage of kg in total catch | Individuals (n) | Percentage of n in total catch |
| *Tilapia* spp. (i) | 1805 | 82 | 53345 | 93 |
| *Channa maculata* (iv) | 192 | 8.8 | 1513 | 2.7 |
| *Carassius auratus auratus* (i) | 73 | 3.3 | 1499 | 2.6 |
| *Cyprinus carpio carpio* (i) | 65 | 3.0 | 539 | 0.9 |
| *Micropterus salmoides* (i) | 43 | 2.0 | 143 | 0.3 |
| *Anguilla* spp. (na) | 9.0 | 0.41 | 24 | 0.04 |
| *Awaous aenofuscus* (na) | 3.0 | 0.13 | 15 | 0.03 |

Distribution status class: i = introduced, iv = invasive, na = native. Sources: [30,37].

### 3.2. The High Popularity of Fishing

Though interviewees stated that "*The fishes were more numerous and easy to catch before*" (VF3), and hence noticed that fisheries have become less profitable (16%), the possibility to catch fish all year round (44%) and the remaining big and/or valuable fishes (27%) gave rise to the recent popularity of fishing as an occupation in the Alaotra region (Q33). The daily income generated (Q14) by fishing, 2.5 USD (*n* = 109, ranging from 0.38 to 12.6 USD), still lies well above the national average of 1.4 USD [38]. However, only 17% (Q15) of the respondents are full-time fishers, earning their livelihood exclusively by fishing, whereas the majority (83%) work part-time as fishers (including seasonal, occasional and part-time fishers). The daily income (Q14) generated by fishing, however, did not differ between full-time and part-time fishers. Further results show that the combinability, low entry and exit barriers, as well as the low opportunity costs of fisheries, are key factors for their popularity: (i) a large part of the interviewees fish for seven days a week (rainy season: 55%, dry season: 60%, Q10 1,2) but hours spent in daily fishing trips (Q9) are not more than 3:42 h on average (see Table S3). (ii) The most commonly used method are traps (60%; Q6), which are usually placed and checked during night and early morning hours, and therefore allow for other occupations during daytime. (iii) Besides the method's easiness (64%, e.g., easy usage and maintenance, non-arduous work) and economic advantages (20%, e.g., high catch rates), its combinability with an additional occupation was a key reason for fishing method selection (12%; Q7): "*I can still do another job after checking the vovo* [*fish trap*]" (VF15). (iv) Likewise, practical and economic reasons were main determinants of occupational choice (Q5) for around half of the fishers, rather than family tradition (25%) and personal aspiration (27%): "Easiness" (20%) was mainly specified as low barriers regarding required education (9%; "*You don't need to be educated*", AGF1; "*I failed at school*", VF24), low special effort (6%; "*I live next to the harbor. That's why I work in the water*", ASF8) as well as little financial investments needed (2%). Economic

factors are also important for becoming a fisherman (need of additional job, 6%; lacking alternatives, 8%; poverty, 2%; year-round source of income, 13%).

*3.3. Perceived Problems and Future Perspectives of the SSIF*

In fact, there was a clear awareness of the decline in fish catches, perceived by 85% of the respondents as the main change in fisheries (Q34). Moreover, 63% of the fishers (Q31) claimed to be personally affected. Other problems the Alaotran fishery sector has to contend with (Q30) are social conflicts and criminal behavior, brought up by 85% of the respondents (Q30). 54% of the fishers claimed to be afflicted (Q31) and reported, for example: "*The people using Tosika [illegal method where fishes are trapped in an enclosure out of plants built on the lake bottom] fish next to the Hamatra [reed walls to attach fish traps] and disturb the fishes. There are also people stealing fishes out of the vovos when they see that there are some* [*fishes*] *inside*" (ANF12). Generally, conflicts were specified as stealing fishing gear or fishes (24% and 14%, respectively) and disturbing or destroying fishing gear (15%). Other problems were lacking equipment, smaller fishes and low water levels.

Questioning about the barriers for becoming a fisherman (Q23) further disclosed a poor motivation to fish if someone had enough money, fields or work on the land (25%) or other jobs (4%), since many respondents misunderstood the question. Interviewees argued that: "*People that have a lot of fields do not have to work in the water*" (ASF4) or that "*They have enough work on the land*" (ANF29). Direct barriers mentioned were money for fishing gear (41%), boats (8%), learning to fish (13%) and the fisherman's hard and frightening work (7%; "*They fear the water*", AGF5; "*Fishing is a hard work*", ASF24).

Fishers' awareness of the aforementioned challenges was reflected by comments on fisheries' future perspectives (Q41), which were generally negative (85%). Only 8% of fishers still believed in the continuity of the Alaotran fisheries. Another group (11 %) offered a disillusioned view with a more or less fatalistic attitude toward fishing, expressed by statements like "*There is no change anymore so we work on land and in the water, even if it is difficult*" (ASF17), or attributed to a perceived uncertainty or lack of alternatives outside the fishery sector: "*The future is unknown but I will go on with fishing because I am used to it*" (VF27). Fishers' negative perspective became apparent in their desperation and the hopelessness they associated with fishing: "*I do not have any hope regarding the livelihood here and I am still searching for solution until now*" (VF2), whereas a quarter of them specified: "*There is no hope anymore for fishing so I have to find other jobs*" (VF5).

Thus, it is not surprising that the clear majority (71%, *n* = 83; Q40) disliked the idea of their sons becoming fishers like themselves. This was justified by the livelihood insecurity nowadays related to fishing (64%), a high burden of work (18%) and the aspiration that their sons might get a "better job" (18%). However, more than one quarter (28%, *n* = 33) still wished for their sons to become fishers, mainly because of adhering to family tradition (52%) and having little other assets (52%): learning to fish was seen as a solution for school failure (18%), shortfalls due to poverty (18%) and as an alternative to unemployment or even criminality (15%) by generating at least a small income. Arguing that "*Like that he can still work but not steal in case he will not find other jobs*" (VF22) and emphasizing that "*Agriculture is now bad so he has to fish since he lives next to the water*" (VF15) showed fisheries function as a safety net for the rural poor.

*3.4. The Expanding Fishery Sector—Fishing as a Rural Adaptation Strategy*

People at Lake Alaotra have been entering the fishing sector for decades. One driver is certainly the daily generated income (2.5 USD), which is at the lower end for the Alaotra region per capita income of 2.5–5 USD [27] but still lies above the national average. However, the investment capacity (Q14) of fishers remains low, since only 21% of the interviewees claimed to have enough income to invest in their business and thereby to improve their economic situation. Half of the interviewees (55%) said they were able to cope with financial bottlenecks, while for one quarter (25%) income was just enough to buy food but not for building up any reserves.

Fishing was mostly combined with former livelihood activities to provide additional income (58%, *n* = 58; Q15B), mainly to complement agriculture (85%). Livelihood diversification was mainly based on economic reasons like lower yields, too few fields to feed the family (including having to share fields with children), decreased income or general livelihood insecurity: "*Life is becoming difficult and the yields are decreasing*" (AGF29). Statements like "*There are not enough fields to use*" (ANF13) and "*I do not have enough field*" (VF16) pointed out the general lack of agricultural land, which hinders interviewees from expanding agriculture. Full-time fishers (17%, *n* = 20; Q15A) largely lamented that fisheries do not support their livelihoods anymore (85%), but a lack of fields, equipment or education prevents them from practicing another occupation. Likewise, former full-time fishers (25%, *n* = 29; Q15C) started to diversify their livelihood, notably during the last decade (70%). Reasons were the declining catches (28%), decreasing income (17%) and livelihood insecurity (14%).

*3.5. Regional Climate and Land-Use Changes—Drivers of Livelihood Diversification?*

97% (*n* = 113; Q38 1,2) of the respondents noticed changes in the Alaotra region, mainly referring to smaller catches (57%), yield reduction in agriculture (12%) and climate variability related topics (16%). As for the latter issue, the interviewees explained that "*The year is disturbed*" (ASF29), "*The rain is delayed*" (VF14) and that "*A lack of rain decreases the water level in the Alaotra*" (ANF6). Some of them further noted that "*[ . . . ] it makes the rice planting late*" (ANF18) and "*[ . . . ] delays the fishing and the agriculture and [makes] the village [ . . . ] insecure*" (ANF23) as well that "*The income is bad because of the low water level*" (ANF24). The increase of social conflicts (9%) and criminal behavior (8%) were additional changes observed. Interviewees reported that "*People in the society are jealous of each other and have conflicts with each other*" (ASF11), that "*Due to the difficulties of life the society is destroyed*" (ASF30) and that "*The insecurity increased a lot in the Alaotra*" (VF10). The few remarks referring to positive changes (5%) were largely premised on better yields or catches due to unsustainable or illegal practices—either fishing during the fishing closure period or the planting of "*vary jebo*". Reasons for change (Q38 3) for the worse (climate variations, environmental degradation, livelihood insecurity, diminishing catches and harvests, use of illegal methods, destruction of small fish, criminal behavior, jealousy, weak government) attest to a strong nexus between environmental, societal, political and economic dimensions and the fishery sector. Local societal linkages seem strongly under stress, as one interviewee observed: "*Even the educated people are fishing because they don't find other jobs*" (AGF21).

Asked about trends in the rainy season (Q36 1,2,3), almost all interviewees mentioned its delay. 98% (*n* = 112) mentioned that this has a direct negative impact on their livelihood, notably fishing (65%, as delayed activities/income and fewer fishes due to a later increase and overall lower water level) and agriculture (32%, referring to delayed activities and poor or delayed yields). Asking fishers about the start of the rainy season in their childhood (Q35) revealed that the older they were, the earlier the rainy season started, according to their memories. 96% of fishers dating their childhood more than 40 years ago (40–59 years, *n* = 26) stated that the rain season had started before December in their childhood, while the same applies for 72% of those dating their childhood over 20 years ago (20–39 years, *n* = 60) and for only 42% of the younger interviewees, dating their childhood within the last two decades (4–19 years ago, *n* = 31).

## 4. Discussion

Small-scale fisheries have served for a long time as a livelihood opportunity with low entry barriers in rural areas next to water bodies, and hence as a safety net especially for the rural poor. However, at Lake Alaotra this "net" seems to have been strained to its breaking point.

*4.1. The Lake Alaotran Fisheries—Symptomatic for Imminent Changes in the Tropics and Subtropics?*

At Lake Alaotra the number of people engaged in inland fisheries has increased sharply since the mid 20th century. The first data, from 1954, register no more than 100 fishers, since fishing does not have a long tradition at Lake Alaotra [36]. This number has risen dramatically during the

last 60 years to a total of app. 12,000 fishers. The Lake Alaotran fisheries reflect the rapid growth of the inland fishery sector in developing countries over the last century, triggered by population growth and limited employment possibilities in rural areas [39]. During the last 20 years, worldwide employment in fisheries has grown faster than the world's population, and faster than employment in traditional agriculture [3]. Our results show that the vast majority of the fishers at Lake Alaotra are part-time fishers (83%), earning their livelihood through various occupations. This implies that the National census method may underestimate the number of fishers, since fishers that fish only temporarily may not call themselves "fishers" [3,40–42]. The group of part-time fishers is especially large in developing countries and inland fisheries, since it represents a crucial strategy for livelihood diversification [9,18,39,43]. In our study, half of all fishers earned their living by agriculture before they started to fish. The decision to fish is based on economic drivers like decreasing yield or income, a lack of agricultural equipment or land and general livelihood insecurity. These drivers are characteristic of the tropics and subtropics, where high demographic growth and an increasingly irregular distribution of precipitation are resulting in soil degradation, landlessness and finally crop shortfalls, reduced yield potential and the loss of useable land [44–47]. In the Inland Niger Delta, fishing has served as a safety net for farmers since droughts in the 70s and 80s have led to decreased harvests [48].

A similar case has been observed in Lake Mweru and the Luapula River fisheries (located on the border of northern Zambia and the Democratic Republic of Congo), where droughts are likely to increase the influx into the fishery sector from agriculture [49]. So, Lake Alaotra seems to reflect an alarming trend, which can already be seen in many regions of the world and may affect a larger number in the near future. This development is contrary to the important role that SSIF should play in reaching sustainable development goals like ending poverty by 2030 [50].

### 4.2. The Immanent Need for Livelihood Diversification—People's Adaptation to Change

Altered conditions for agriculture and lacking alternative sources of income are the main reasons for the expansion of the inland fishery sector at Lake Alaotra. Supplying one third of the country's rice production, the Alaotra region became the so-called "rice granary" of Madagascar and therefore a favored region for migration [26]. Today, the rising need for agricultural land diminishes the remaining wetlands due to their conversion into rice fields, while the intensification of rainfed agriculture has led to soil fertility loss and erosion. Low investment capacities, fluctuating prices and the high intra- and inter-annual changes of the rainy season expose local farmers to additional risks. As an overall result of those interrelated changes, yields in crop production have declined or at least stagnated [32,48,49,51–53]. On an individual level the proportion of agricultural land per person (or rather family) is further reduced because fields are shared within the next generation. Decreasing yields and lack of agricultural land were the main reasons that led a large part of fishers to start fishing on a part-time basis. Interlinked drivers (e.g., decreasing income, lacking equipment and livelihood insecurity) show that the higher population density implies a rising competition for employment opportunities. Welcomme [54] concludes that part-time fishers, earning income from other sources (e.g., labor, transport), will invest less time in fishing as soon as it proves unprofitable. However, as the author adds, part-time fishers whose alternative sources of income are on a seasonal basis (e.g., seasonal crop cultivation), will continue to fish during downtimes, even when catches are low. For Lake Alaotra, cultivation downtimes determine the annual life cycle of the local population, as agriculture is dictated by the crop seasons [55]. During those downtimes, fishing is often the only alternative source of income. We assume that the entry into fisheries at Lake Alaotra will be increasingly guided by economic reasons, their open access nature and their easy practicability, rather than tradition and personal preference: at least half of the interviewees started to fish because of livelihood insecurity, the opportunity of cash income on a daily basis and the low entry barriers of fishing (no need for formal education and low or no need for financial investments). Since our study disclosed that more than two thirds of the interviewees wish their sons to pursue a different profession, the lack of alternatives is one important reason for the sector's continuing popularity.

A similar trend was observed by Morand, Sy and Breuil [56] for West Africa, where the livelihoods of rural and peri-urban people are increasingly dependent on fishing. The increased need for cash, the lack of alternative sources of income and a consistently increasing demand for fish are the main reasons encouraging people to take up fishing. They emphasize the increasing shortage of farmland, the unpredictable nature and low profitability of rain-fed agriculture under the increasing variability of climate, the low work possibilities for rural people in cities due to educational deficits and the insecure livelihoods in the informal sector. Considering the trend of a progressively shortening of the rainy season at Lake Alaotra [57] and of a "shifting baseline" pattern regarding the onset of the rainy season in our study, economic shortfalls in other sectors will accelerate the influx in inland fisheries, as it is seen to be one of the major indirect impacts of climate change on fisheries [58]. The impact of such a scenario on the local population of fishers is nevertheless poorly analyzed [59].

*4.3. Constraints and Consequences of Development—From a Safety Net to a Poverty Trap*

The alternating labor transfer between agricultural and fishery sectors is common in rural areas that are poor in resources and close to water bodies [4,59]. However, at Lake Alaotra, a scenario becomes apparent where both sectors seem to have reached their upper limit of expansion and where fisheries might lose their crucial "safety net" and "labor buffer" function, as it is defined by Béné [60]. Those mechanisms are "*crucial from a social and economic point of view, especially in remote areas where alternative employment may be scarce and social-security programmes either minimal or nonexistent*" [60], and they have protected people from economic crises, political conflicts and natural catastrophes worldwide and given a livelihood to the rural poor (e.g., job losses [61]; civil war [60]; weather events [62]; agricultural failure [63]; food insecurity [64]). At Lake Alaotra, agriculture and fisheries are suffering substantial losses simultaneously, driving farmers to engage in fishing and fishers in farming. Lacking alternatives prevent them from leaving, trapping them instead in declining sectors. A point of no return seems to be indicated by the rising number of fishers, even in light of (i) fishers' overall negative perspective on the future of fisheries (ii) the unpopular idea of their sons becoming fishers and following family traditions and (iii) the variety of challenges fishers are facing (e.g., declining catches, livelihood insecurity, high workload, criminal and adversarial behavior, low water levels). The fact that (iv) current full-time fishers are willing to exit fisheries but hampered due to lacking assets (fields, money for equipment, education) supports the hypothesis according to which fisheries are reaching a point of no return and becoming a trap rather than a safety net.

Current developments are reflected in a sustained annual catch decline (reduced to less than one quarter over the past 50 years) as well as changes in species composition [22,36,65]. Lower reproduction rates, since individuals are largely caught before reaching maturity [66,67], genetic deterioration and trophic composition changes are most likely involved. Additional interlinked effects and cross-sectoral responses linked to overfishing are increasing conflicts and competition over livelihood opportunities, fishing places and gear, decreasing income and increasing poverty. As observed in other regions [68,69], local people balance livelihood insecurity by usage of destructive fishing methods and the unsustainable expansion of agriculture into the marshes, leading to the destruction of spawning sites and criminal behavior (e.g., stealing). According to Pomeroy [70], who documented such a negative feedback cycle in Indonesia, this occurs especially when rapid population growth, fewer livelihood opportunities and poor access to land intensify fishing pressure and can lead to fish stock depletion or collapse and food insecurity, which can have tremendous impacts on human behavior, social environments as well as ecosystem properties. Moreover, such an undesired development is leading to an overall increased vulnerability of communities [39].

The fishers' willingness and ability to abandon fishing will play a crucial role in the long-term sustainable exploitation of stocks. Following Daw et al. [71], the entry into and exit from fisheries is not always guided by profitability, as many models suggest. The local and individual context (e.g., socio-economic factors, age, education, occupational attachment and identity, type of fishing) is assumed to be significant for exiting and therefore may force fishers to stay in fisheries although

profitability is low [71]. Smith [43] and Martin [72] showed that fishing, used as a strategy to spread risks, is often maintained alongside other occupations to keep economic mobility, regardless of opportunity costs, whereas Cinner [73] and Daw [71] concluded that the availability of alternative livelihood options increases the opportunity costs of, and enhances the displacement from, fisheries. This implies that regions lacking alternative livelihoods, as the Alaotra region, are more affected by overfishing. The prevalent use of passive fishing methods during nighttime (making fishing combinable with daytime labor) and low-cost gear (often self-made traps) indicates the already low investment capacities (either time and/or money) of the Alaotran households [43]. Decreasing productivity in fisheries and agriculture will lower investment capacities even further and make displacement in other sectors increasingly unlikely.

## 4.4. Future Development Considerations

Our findings confirm that the safety net function of SSIF is crucial for rural people's livelihood security and a sound social environment. Secondly, our study indicates that current trends of population growth, land use change and climate variability in developing countries may strain this safety net to its breaking point: increasing human pressure may exceed fish stock regeneration capacity and let people "fall" into a poverty trap [73]. In line with conclusions of the literature about poverty traps [73,74], Alaotran full-time fishers were largely excluded from alternative livelihood strategies with higher returns, while at the same time lacking alternatives force other people to enter fisheries despite their declining productivity. Although a large amount of literature has emerged about the importance of small-scale fisheries' welfare function and their policy implications [16,74,75], policy objectives mainly adhere to mainstream wealth-based models focusing on maximizing economic rent and gross domestic product contribution while explaining poverty by the Malthusian logic [13]. As policies aligned with wealth-based models focus on resource access control and fishing effort reduction in order to increase fisheries' overall productivity, they run the risk of excluding precisely those rural poor whose livelihoods depend on it [16]. The case of the Alaotran fisheries shows that a more profound analysis of people's livelihoods can identify ongoing livelihood dynamics and disclose possible poverty trap mechanisms. Awareness of the dichotomy between the welfare and the wealth-based models underlying policy and an understanding of fisheries' current functions are essential for local management orientation and the understanding of management failures: (i) spatial fishing closures combined with gear restrictions can be more effective than temporal fishing closures if fishers are highly dependent on daily cash income [30,58,76]; (ii) gear exchanges for fishers practicing illegal and destructive methods can encourage compliance with gear restrictions and reduce social tension [76]); (iii) temporal fishing closures should consider fish species' reproductive cycles and local livelihood adaptations to intra-annual cycles. The current fishing closure at Lake Alaotra is too late to effectively protect spawning fishes [25]. The closure aims to encompass the rice-cultivation period, with its high employment opportunities outside fisheries, but Ducrot and Capillon [55] have shown that the time of crop establishment is the most critical time for farmers. During this time, when cash is scarce and hungry gaps can hardly be avoided, local populations rely mostly on the additional daily income provided by fishing. (iv) Management should foster dynamic (temporal) or mobile (spatial) fishing closures for a more equal distribution of benefits and costs [25], facilitate a flexible reaction on the inter-annual changes and shifts by climate variability and consider spatial and temporal trade-offs between land use and fisheries, especially with regard to current precipitation patterns [60]). (v) Economic mobility is required to overcome shocks and to adapt to change. Income decline can cause stochastic (often temporary) poverty, while the loss of assets leads to structural (often persistent) poverty, where economic mobility is lost and people may be caught in a poverty trap [77]. Management interventions should therefore also focus on building and protecting fishers' assets [15,74,77].

## 5. Conclusions

SSIF still offer a much-needed opportunity for livelihood diversification for local populations near inland waters facing livelihood stress worldwide. Such opportunities are in rising demand with the increase of external pressures like climate change or disruptive pandemics, such as the current global spread of the Covid-19 virus, or internal stressors like population growth, overexploitation of natural resources and economic crises. Our study from Lake Alaotra, Madagascar, shows that SSIF's function as a safety net may already have been stressed to a breaking point. Therefore, its current potential for alleviating poverty may increasingly become a poverty trap, that still allows people to enter the SSIF, but with no way out. Being caught in a poverty trap will lead to increasing overexploitation, further undermining SSIF's safety-net function. The scenario from Lake Alaotra may well reflect the future prospect of many SSIF worldwide, and can therefore be seen as a warning. Policy makers and international agencies are now obligated to fathom the local value of fisheries' welfare function and decide about the implications for future fisheries management.

**Supplementary Materials:** The following are available online at http://www.mdpi.com/2071-1050/12/18/7299/s1, Table S1. Interviews performed with the Alaotran fishermen. Interviews comprise 41 questions and were performed with 117 fishermen at Lake Alaotra. Table S2. Results from interviews with fishers. Results from the interviews with Alaotran fishers (*n* = 117) showing the number of respondents and the percentage (%) of respondents for each question. Table S3. Fishing trip duration. Median time for travel, fishing and total trip (hh:mm) indicated by Alaotran fishers from Andilana Sud, Anororo, Vohimarina and Andreba for the dry season (DS) and wet season (WS). Fishing times between seasons were compared with Mann–Whitney U tests.

**Author Contributions:** P.L.L., T.R. and J.M.-C. conceived and designed the study; P.L.L. conducted the field work and interviews, and analyzed the data; P.L.L., T.R. and J.M.-C. prepared the manuscript. All authors have read and agreed to the published version of the manuscript.

**Funding:** This research was funded by the Bauer-Foundation of the "Deutsches Stiftungszentrum", grant number T237/22985/2012/kg.

**Acknowledgments:** We thank all participants of this study and particularly Lala Nomenjanahary Elysé and Bernard Aimé Rajaonarivelo for their assistance with the field work. We also thank the community of Andreba, Andilana Sud, Anororo and Vohimarina for their cooperation and hospitality, as well as Durrell Wildlife Conservation Trust in Ambatondrazaka, the Service Régional de la Pêche et des Ressources Halieutiques and the Ministère de l' Environnement, de l' Ecologie et des Forêts for supporting this work.

**Conflicts of Interest:** The authors declare no conflict of interest. The founding sponsors had no role in the design of the study; in the collection, analyses, or interpretation of data; in the writing of the manuscript, and in the decision to publish the results.

**Ethics Statement:** Field work permits were issued by the Madagascar's Department of Environment, Ecology and Forest. Fishers declared to be available and willing to participate and were informed that their identities and responses would not be shared with anyone. Ethics approval for the study was granted by the Research Ethics Committee of the University of Hildesheim.

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
