# Peer review of "From Safety Net to Point of No Return—Are Small-Scale Inland Fisheries Reaching Their Limits?"

_sustainability, doi:10.3390/su12187299_

Round 1

Reviewer 1 Report

This is a very interesting and well written article about fisheries in poor and developing countries. I read this text with great pleasure, having in my minds my own experience from those countries in Africa.

  1. I think that in the general part (line 39) authors could mention the pollution in these countries. Pollution of the natural environment, including rivers, flowing into the ocean, and pollution of the coastal zone are pivotal problem in African countries. Nowadays many fishes, birds in those parts of worlds are pollutant by plastic microfibers.
  2. Maybe you can show some results in the main part of the text on the charts? It will be easier to see the results.
  3. You can highlights the conclusion
  4. You can add a few sentences as limitation of your research in separate section.

Line 21 … decreasing agricultural yields force farmers into livelihood diversification – earlier sentences speak of fishing, so agricultural yields and diversification in agriculture is a sentence incomprehensible in this context. Please elaborate this thought or move this sentence in other place of abstract.

Line 39 growing demand, environmental degradation and climate change – environmental degradation is really huge problem for Africa countries. Maybe you can add 1-2 sentences about pollution of water (river tributary to the see with tones of plastic waste, seaside) and impact on fishes (e.g. https://doi.org/10.1515/oszn-2017-0026 or DOI: 10.1016/j.envpol.2017.07.103)

Line 99 This is description of region, next you made description of Madagascar. So the sentence “Main sources of income are SSIF and rice cultivation” fits the previous context, which is the whole of Madagascar, and probably only applies to the region?

Line 202 This is a very important Figure. I think that it would be worth extending the measurement period to 2018 or 2019. The authors ended in 2011 (I read explanation in line 199) and if these lines were to be extrapolated, the catch would be at 0. Please try to find the latest statistics data (of course if possible). If not possible, please make your own simulation of the trends.

Line 228 From the management point of view, maybe you can add exit barriers as well. If exit barriers are also low (please confirm if it is truth in your case), sector entry is more popular than if exit barriers were high.

Line 335 Please add the country for those Lakes

Line 384 and Line 387 Please make a better joint/transition of that information. Your research was in East Africa (Island) and the next sentence is about West Africa. At this moment there is no logical transition

Author Response

Authors: First of all, we would like to thank you for reviewing our manuscript. Your comment was plausible and helpful. We hope the following changes will remove your remaining concern.

  1. I think that in the general part (line 39) authors could mention the pollution in these countries. Pollution of the natural environment, including rivers, flowing into the ocean, and pollution of the coastal zone are pivotal problem in African countries. Nowadays many fishes, birds in those parts of worlds are pollutant by plastic microfibers.

Authors: We have added pollution as a problem in the introduction (line 39)

  1. Maybe you can show some results in the main part of the text on the charts? It will be easier to see the results.

Authors: We understand your point. However, as you can see in the table “Results from interviews with fishers” (supplementary material) the main results are very comprehensive. Even if we join just some of the main results in a chart, it would be quite big and therefore rather complex than giving an overview. Therefore, we decided to mention the results in the text and add the whole table to the supplementary material. In addition, we wanted to keep the manuscript as short as possible

  1. You can highlights the conclusion

Authors: Thank you for this comment. We added a conclusion at the end of the manuscript to highlight the most important results and to outline future prospect of SSIF worldwide.

  1. You can add a few sentences as limitation of your research in separate section.

Authors: We included a sentence explaining the trend until 2016. However, this data is no official data and therefore not shown in the figure.

Line 21 … decreasing agricultural yields force farmers into livelihood diversification – earlier sentences speak of fishing, so agricultural yields and diversification in agriculture is a sentence incomprehensible in this context. Please elaborate this thought or move this sentence in other place of abstract.

Authors: We added more information to this sentence and hope that our point is understandable now: ”A point of no return seems near as decreasing agricultural yields force farmers to enter the fishery sector as a livelihood diversification.” (Line 21/22)

Line 39 growing demand, environmental degradation and climate change – environmental degradation is really huge problem for Africa countries. Maybe you can add 1-2 sentences about pollution of water (river tributary to the see with tones of plastic waste, seaside) and impact on fishes (e.g. https://doi.org/10.1515/oszn-2017-0026 or DOI: 10.1016/j.envpol.2017.07.103)

Authors: We added a short note on the importance of water pollution and appropriate sources: “However, SSIF, as fisheries in general, are increasingly facing challenges, e.g. growing demand, environmental degradation, especially water pollution and climate change….” (line 39)

Line 99 This is description of region, next you made description of Madagascar. So the sentence “Main sources of income are SSIF and rice cultivation” fits the previous context, which is the whole of Madagascar, and probably only applies to the region?

Authors: Thanks for pointing that out. We´ve changed it. (line 110/111)

Line 202 This is a very important Figure. I think that it would be worth extending the measurement period to 2018 or 2019. The authors ended in 2011 (I read explanation in line 199) and if these lines were to be extrapolated, the catch would be at 0. Please try to find the latest statistics data (of course if possible). If not possible, please make your own simulation of the trends.

Authors: Since the data collection is very unregular in the study area and since we got my statistics directly from a governmental authority, we have no additional data on current years. There are no new data on the number of fishermen. However, we´ve some data on fish catches up to 2016. We added an sentence about the trend shown by those additional data which were added to the official data handwritten (…and therefore indicated as “non-official”): “Non-official data on fish catches up to 2016 indicate a stabilization at the low level reached in 2011.” (line 212/213)

Line 228 From the management point of view, maybe you can add exit barriers as well. If exit barriers are also low (please confirm if it is truth in your case), sector entry is more popular than if exit barriers were high.

Authors: Since the entry barriers for the fishery at Lake Alaotra are low (no need of formal education and low or no need of financial investments necessary) there exist no exit barriers. We added this fact in the sentence in line 240/241 (formerly line 228): “Further results reflect that the combinability, low entry and exit barriers as well as low opportunity costs of fisheries are key factors for its popularity:…”

Line 335 Please add the country for those Lakes                                                                                   

Authors: We suppose you are referring to Line 356. We added the country for the mentioned lake and river. (line 367/368)

Line 384 and Line 387 Please make a better joint/transition of that information. Your research was in East Africa (Island) and the next sentence is about West Africa. At this moment there is no logical transition.

Authors: We added the transition: “A similar trend was observed by Morand, Sy and Breuil [56] for West Africa where livelihoods of rural and peri-urban people increasingly rely on fishing. The increased need for cash, the lack of alternative sources of income and a consistently increasing demand for fish are the main reasons encouraging people to take up fishing.” (line 399/401)

Reviewer 2 Report

The manuscript “From Safety Net to Point of no Return – Are Small- Scale Inland Fisheries Reaching Their Limits?” is very interesting. However, it requires additional technical explanations to improve the clarity for readers:

INTRODUCTION

It should show a wider literature review, not only of the local aspect of the problem. This should be presented as a global problem with a particular local example.

  1. Materials and Methods

Subsection 2.1 – it cannot be a subsection but an information placed in the end.

…” 2.1. Ethics statement

Field work permits were issued by the Madagascar’s Department of Environment, Ecology and Forest. Fishers were questioned as available and willing to participate and were informed that their identities and responses would not be shared with anyone. Ethics approval….”

- Subsections 2.2. Study site & 2.3. Lake Alaotran fisheries – should be merged into one subsection, e.g. Study area. The data on lakes and fish species should be placed in tables for readability (they are then easier to come back to). There could potentially be two subsections within the section Study area. The maps should be placed at the beginning of the section.

- Subsection 2.4. Regional changes and threats – please remove. The information from it may be moved to INTRODUCTION and Study area.

  1. Results

3.1. Evolution of the Alaotran fisheries

Table 1

Percentage (%) – why the accuracy is given once as 79.6 (three significant digits) and then as 17 (two significant digits), then again 3 (one significant digit)?

Further in the table, the presented data also have a different number of  significant digits. Why?

Catch weight (kg) – from 1 to 3?

Percentage of kg in total catch – from 2 to 3?

Percentage of n in total catch – from 1 to 3?

There should also be an extra section added: CONCLUSION.

The subsection 4.4 Future development considerations – this is very interesting but only local  description.

The summary should show the influence of the obtained results on similar problems worldwide.

Author Response

Authors: Thank you for your feedback and valuable comments. They helped us to improve our manuscript and to eliminate incomprehensibilities.

INTRODUCTION

It should show a wider literature review, not only of the local aspect of the problem. This should be presented as a global problem with a particular local example.

Authors: We think we made a clear point of highlighting the worldwide dimension of small scale fisheries, especially inland fisheries, and the problems they are facing (Source 1-24). We think that we clearly presented our study as an example placed in a greater context. So we disagree with your point that the manuscript would benefit from further extending the cited literature, it would perhaps even lose in readability and focus.

  1. Materials and Methods

Subsection 2.1 – it cannot be a subsection but an information placed in the end.

…” 2.1. Ethics statement

Field work permits were issued by the Madagascar’s Department of Environment, Ecology and Forest. Fishers were questioned as available and willing to participate and were informed that their identities and responses would not be shared with anyone. Ethics approval….”

Authors: We agree and moved the subsection 2.1 to the end of the manuscript. The reader will find this information after the information about “Author Contributions” and “Funding”.

- Subsections 2.2. Study site & 2.3. Lake Alaotran fisheries – should be merged into one subsection, e.g. Study area. The data on lakes and fish species should be placed in tables for readability (they are then easier to come back to). There could potentially be two subsections within the section Study area. The maps should be placed at the beginning of the section.

- Subsection 2.4. Regional changes and threats – please remove. The information from it may be moved to INTRODUCTION and Study area.

Authors: We placed the map at the beginning of the method part and merged the first two sections (2.2. Study site & 2.3. Lake Alaotran fisheries). Further we replaced the part about the fish species by a new table (Table 1) and included the part 2.4. Regional changes and threats to the Introduction.

  1. Results

3.1. Evolution of the Alaotran fisheries

Table 1

Percentage (%) – why the accuracy is given once as 79.6 (three significant digits) and then as 17 (two significant digits), then again 3 (one significant digit)?

Further in the table, the presented data also have a different number of significant digits. Why?

Catch weight (kg) – from 1 to 3?

Percentage of kg in total catch – from 2 to 3?

Percentage of n in total catch – from 1 to 3?

Authors: Thanks for that helpful comment. It has made us realize, that the table in the manuscript was added in with wrong formatting. That’s probably why the numbers were accidentally shown wrong. In the new table two-digit numbers have no decimal place, one-digit number have one decimal place and numbers below zero have two decimal places. This counts for all data (individuals, data….). You can see the new table below:

There should also be an extra section added: CONCLUSION.

The subsection 4.4 Future development considerations – this is very interesting but only local description.

The summary should show the influence of the obtained results on similar problems worldwide.

Authors: In the last section “Future development considerations” we describe solutions approaches which apply for the Alaotra region but also for other regions with similar problems worldwide.  Those recommendations are numbered with (i), (ii), ...(v).

However, we added a conclusion to give a short overview about the most important results and the future prospect of SSIF worldwide. Thank you for this comment.